# Caudal Regression Syndrome—A Narrative Review: An Orthopedic Point of View

**DOI:** 10.3390/children10030589

**Published:** 2023-03-19

**Authors:** Barbara Jasiewicz, Wojciech Kacki

**Affiliations:** Department of Orthopedics and Rehabilitation, Jagiellonian University Collegium Medicum, Balzera 15, 34-500 Zakopane, Poland

**Keywords:** caudal regression syndrome, sacral agenesis, pediatric spine

## Abstract

Abnormalities in cellular differentiation during embryo-fetal period may lead to various malformations of the spine. Caudal regression syndrome (CRS) is a group of defects with premature growth/development termination of the vertebral column. CRS can be divided into three types: sirenomelia, complete absence of the sacrum and partial absence of the sacrum. Genitourinary and gastrointestinal anomalies are common, with neurogenic bladder and bowel incontinence. Treatment of patients with CRS is complex and multidisciplinary and should be comprehensive. The most common orthopedic problems are: spinal deformity (kyphosis and scoliosis), spinopelvic instability and lower limbs deformities.

## 1. Introduction 

Abnormalities in cellular differentiation during embryo-fetal period may lead to various malformations of the spine [1]. Caudal regression syndrome (CRS) is a group of defects with premature growth/development termination of the vertebral column [2]. The extent of the defect may vary from partial to complete absence of the sacrum and/or distal lumbar vertebrae. The CRT is also known as caudal dysplasia sequence, sacral regression syndrome, and a large number of authors refer to it as sacral agenesis [3].

In embryonic period, before the fourth week of gestation, there is a developmental arrest of the caudal mesoderm [4,5,6]. This failure affects the midposterior axis of the mesoderm. The proximity of caudal neurons, spinal, hindgut and other elements involved in the closure of the neural tube results in a set of defects in various systems [5]. These abnormalities are often quite complex with spinal defects accompanied by numerous anomalies of the viscera including anorectal region of the gastro–intestinal tract, genitourinary systems as well as neurological deficits of varying severity—from urinary incontinence to complete paralysis of the lower limbs [1,2,4,7,8].

Caudal regression syndrome is a rare disorder, with an incidence of about 1–2:100,000 according to some authors, or 1:60,000 live births according to others [3,6,9,10]. This defect is more common in boys, with an M:F ratio of 2.7:1, but in a recent review of 83 cases, there was no gender predilection [3,10]. The most common CRS defect is sacral agenesis (partial or total), with an incidence of less than 0.5% [11]. The exact prevalence of CRS may be difficult to determine, as the mildest forms of the defect, lacking only the coccyx, may be completely asymptomatic or escape detection [7,12].

## 2. Material and Methods

An extensive search of the PubMed and Google Scholar databases was performed up to 10 March 2023 using the following keywords: (“caudal regression syndrome”), (“sacral agenesis”), (“sirenomelia”), (“caudal regression syndrome” AND “treatment”), (“caudal regression syndrome” AND “surgery”), (“sacral agenesis” AND “treatment”), (“sacral agenesis” AND “surgery”), (“caudal regression syndrome” AND “spine”), (“sacral agenesis” AND “spine”), (“caudal agenesis”) and (“caudal agenesis” AND “treatment”). A thorough search of published literature resulted in more than 3000 relevant articles presenting cases and reviews of CRS (Table 1). A snowball technique was applied in search for applicable references of eligible studies and reviews.

Regarding the study design, the cohort, case–control, cross sectional, case reports and case series were selected without gender, language or other demographic bias.

The selection of studies was performed by the authors using following criteria:Recognized key publications in the history of CRS studies;Studies fully describing the etiology/epidemiology/morphology of the spinal defect;Studies of interest from the orthopedic point of view (both authors are orthopedic surgeons);The latest treatment solutions.

Only full text studies were considered, either in the language understandable by the authors (English, Polish, German, French) or in others with full translation.

The final choice of the publications selected for further analysis was performed by both authors working independently; any possible disagreement was discussed and resolved through reviewers and team consensus.

According to above criteria, a group of 81 papers was chosen for review.

## 3. Etiology and Genetic Background

The etiology of CRS remains unknown, although both observational and genetic studies shed some light on the problem [2,11,13]. Genetic and environmental factors play a crucial role in CRS pathobiology [14]. Patients with CRS basically can be divided into two groups: the first with a maternal diabetic tendency and the second with a genetic predisposition [15].

The incidence of the defect increases significantly in mothers with diabetes, and Chan reported that about 1% of children born of diabetic mothers have defects from this group [2,16,17,18]. He suggested “increased susceptibility to environmental teratogens during diabetic pregnancy” [2]. Nievelstein emphasized that 16% of children with CRS had mothers who suffered from gestational diabetes [18,19]. Dysregulation of retinoic acid homeostasis may also contribute to the appearance of CRS defects [2,14]. The reasons for an increase in sirenomelia prevalence among young mothers (under 20) are unclear, as well as among monozygotic twin pregnancies [20,21].

Genetic investigations in both human and murine tissue model in vitro and in vivo indicate different genes associated with caudal differentiation in embryogenesis [14]. Studies taking into account the complex and diverse phenotype of CRS patients suggest a multigenic model [22]. It was confirmed that the caudal type homeobox 2 (CDX2) gene plays an important role in caudal morphogenesis and its pathogenic variants can cause CRS defects [13]. CDX1, on the other hand, is important for the development of the caudal region of embryo in studies on mice [23]. Studies of patients with anorectal malformations suggested that downregulation of CDX1 may also be the cause of these defects in humans [24]. It is possible that morphogenesis disorders affecting the spine and the distal gastrointestinal tract may have a similar (or the same) genetic background since both often occur together. CRS was observed in several congenital syndromes, e.g., Currarino syndrome, VACTERL (vertebral defects, anal atresia, cardiac defects, tracheo-esophageal fistula, renal anomalies, and limb abnormalities) or OIES syndrome (omphalocele-exstrophy of the cloaca-imperforate anus-spinal defects) [10,25,26,27]. In OIES syndrome, a defect involving the intraembryonic mesoderm is suspected, with a possible dependence on mutations within the homeobox genes, such as HLXB9 [26]. Although the former investigations suggested a relationship between the HLXB9 gene and Currarino syndrome—the form of CRS with a triad of sacral agenesis (hemisacrum), presacral mass and anorectal malformation [28,29,30]—a further study by Merello et al. proved that the HLXB9 gene is not involved in the pathogenesis of CRS, but is a causative gene only in Currarino syndrome [31]. Cytochrome gene CYP26A1 appears to be more promising: a study examining such single nucleotide polymorphisms and consequent allele variation has found that F186L and C358R variants represent coding regions with products associated with severe impairment in retinoic acid catabolism, a probable contributor to the CRS phenotype [32]. Retinoic acid metabolism plays an important role in morphogenesis, its dysregulation in mice may lead to CRS with spectrum of clinical effects [14,33] The inheritance of other form of caudal regression/dysgenesis syndrome, a sacral defect with anterior meningocele (SDAM), is autosomal dominant [34].

With recent progress in our understanding of embryogenesis, we know more about somitogenesis and the crucial role of MBTPS1/SKI-1/S1P (membrane bound transcription factor protease, subtilisin kexin isozyme-1, or site 1 protease). Conditional Mbtps1 loss-of-function mouse model exhibited phenotypic changes confined to the lumbar/sacral vertebral region, which may mimic those in caudal regression syndrome [35].

Other research proved that PLZF (Promyelocytic leukemia zinc finger protein gene)-deficient rats are affected by the impaired development of the caudal half of the body—caudal regression syndrome [36]. In humans, a rare biallelic mutation of the PLZF gene was described that is similar to caudal regression traits observed in PLZF-deficient mouse and rat strains [37].

Porsch et al. conducted whole-exome sequencing and copy number variation (CNV) analysis human studies in progeny of CRS. This study identified shared mutations in a number of genes, including SPTBN5, MORN1, ZNF330, CLTCL1 and PDZD227 [22]. There are shared genetic variations in multiple clinical CRS subtypes as well as potential overlapping genotypes between VACTERL and CRS [14].

The recent study reported a possible association between ID1 (inhibitor of DNA Binding 1) and non-syndromic sacral agenesis: the missense variants in ID1 were identified in two of three children (paternally inherited) [38]. Future studies are, however, necessary. Most cases seem sporadic, while familial occurrence was reported in some cases, suggesting a possibility of autosomal recessive inheritance [12,32]. In summary, pathogenesis still remains elusive; however, many studies underlined the polygenetic nature of the disease and the influence of environmental risk factors [14].

Multiple anomalies seen in CRS can be explained by the complex embryological process of secondary neurulation. The caudal cell mass (CCM) (an undifferentiated cell mass in the area of the primitive streak) plays the main role in secondary neurulation [32]. The CCM is not only involved in the formation of the spinal cord and the vertebral bodies in caudal area, but is also involved in the formation of surrounding structures (genitourinary anorectal). The attenuation of bone morphogenetic protein signaling at the posterior primitive streak of embryos leads to the caudal dysmorphogenesis (anorectal anomalies and fusion of both hindlimbs) [39]. In his paper, Suzuki underlined an “existence of developmental programs for the coordinated organogenesis of urogenital/reproductive tissues based on growth factor function and crosstalk” [39].

## 4. Classifications History and Present

Historically, the earliest described form of CRS is sirenomelia. In Greek mythology, the image of mermaids has been present for centuries, and the first objective description of sirenomelia appeared in the mid-16th century, by Rocheus in 1542 and Palfyn in 1553 [40]. A more detailed description of this form of CRS, with division into three variants, appeared in the middle of the 19th century (Saint-Hilaire, 1836 and Forster, 1861) [20]. The modern era of CRS begins in the 1960s with the work of Duhamel, who first coined the name “the syndrome of caudal regression” [4]. Despite debate over the years as to whether CRS and sirenomelia are separate entities or one common group of conditions; now, with greater understanding of embryonic development, sirenomelia is believed to be the most severe form of the caudal regression spectrum [6,41,42].

Searching the MEDLINE-PUBMED database, the keywords “caudal regression syndrome” yielded 395 results, which were mostly case reports. Searching for the keyword “sirenomelia” yielded 2588 papers, and most of them began with the words: “a case of sirenomelia is presented…”.

Duhamel, in his 1961 paper, divided CRS into two types: mermaid with lower extremities fused, and anchipodal type with lower limbs flexed in knee joints, abducted in hip joints, with typical popliteal webbing [1,4]. Typical phenotypic appearance of sirenomelia is the presence of axially positioned, single lower limb [20]. Basically, the best and simplest definition of sirenomelia was given by Stevenson in 2006 as “a limb anomaly in which the normally paired lower limbs are replaced by a single midline limb” [21]. Gastrointestinal and genitourinary anomalies usually accompany bone defects.

Depending on the amount of bone elements in the lower limb, Stocker and Heifetz divided sirenomelia into seven types [21]. It was found, over time, that this is not a perfect classification, since there are cases of children who cannot be included in it [43]. Despite this, it is the most popular and frequently used breakdown of the defect, apart from the historical classification of Saint-Hilaire and Foster. Kjaer et al. noticed a relationship between the iliac-sacral distance (ISD) and the severity of the defect (iliac/femur phenotype). However, the proposed division of the defect depending on the ISD value (normal ISD, mildly increased and greatly increased) was not found to be widely used [44].

A general classification of defects from the CRS group was proposed by Welch and Aterman [15]. They divided caudal defects into four clinical groups: three familial types and a non-familial one (often with maternal diabetes) [15].

Congenital anomalies in CRS involve sacrum, coccyx and the lumbar spine, caudal segments of spinal cord and lower limbs. The popular, current classification divides CRS into two groups: first with blunt spinal cord termination above L1 and the second, less severe, dysgenesis with tethered cord [5]. This approach has the greatest clinical implications related to a different prognosis in dysgenetic lumbosacral vertebrae and abnormal distal spinal cord.

Renshaw published a classic CRS classification, dividing it into four subtypes [45]. Types 1 and 2 include various degrees of partial or complete hypoplasia of the sacrum, while types 3 and 4 are the “classic image” of CRS with the absence of some lumbar vertebrae. Some add to this classification type 5—sirenomelia [41].

Stanley et al. combined all sacral anomalies (including meningomyelocoele) and suggested division into three types: agenetic, dysgenesis and dysraphic types [46].

The last, but one worth considering, is the division proposed by Pang which depended on the amount of remaining sacrum and articulation between pelvis and spine [7,12]. It ranged from the most severe to the mildest: type 1 is total sacral agenesis with lumbar vertebral agenesis and type 5 is coccygeal agenesis [7].

Much simpler is the classification proposed by Guille et al., concerning three types of patients and based on their ambulatory potential [47]. Type A includes defects with either a slight gap between the ilia or with their total fusion in the midline along with the absence of one or more lumbar vertebrae. In this group, the caudad aspect of the spine articulated with the pelvis in the midline, maintaining its vertical alignment.

In type B, the defect included the fused ilia, absent some of the lumbar vertebrae, and the most caudal lumbar vertebra articulated with one of the ilia while the most caudad aspect of the spine shifted away from the midline.

Type C is with total agenesis of the lumbar spine, ilia are fused and there is a visible gap between the most caudal intact thoracic vertebra and the pelvis [47].

The above attempts at classifications show the variety of congenital defects classified as CRS. In everyday clinical practice, the most useful classification would provide us with prognostic value and assist in planning further management of those patients. Therefore, the last proposed division, supplemented by MR image of the location of the conus medullaris [12], are the closest to this task. In the latter, in group 1, the conus is absent and the spinal cord ends with a blunted appearance, cranial to the lower border of the L1 vertebra. This image corresponds with a complete absence of the sacrum, i.e., types I and II according to Pang.

In the second group, there is a tethered cord with the conus present below L1. This group includes patients with a partial absence of the sacrum. The increased incident of caudal spinal cord malformations among patients with CRS, commonly described as tethering lesions, as well as the growth and traction, or pressure, on abnormally positioned sacral roots explains the fact that, in some patients, neurologic deficits can be progressive [7,11].

## 5. Morphology Symptoms

Simplifying the classifications according to Pang, Renshaw and Guille, CRS can be divided into three types: sirenomelia, complete absence of the sacrum and partial absence of the sacrum.

The most severe form of CRS is **sirenomelia**. It is often described in human fetuses (premature births, stillborn children) or children who died in the first days of life due to urinary tract defects. Exceptionally, children with sirenomelia are able to survive for more than a year, mainly due to urogenital and anus surgeries. The essence of the defect is a single axial positioned lower limb, with one or two feet (Figure 1) [48]. The bones of the lower limbs may be double or single, and it is significant only for the description/classification.

This anomaly is usually accompanied by defects of the genitourinary organs and the gastrointestinal tract, with the imperforate anus and renal agenesis at the forefront. Malformations of other organs are also common, from hydrocephalus to heart defects and visceral anomalies [20,21]. Prognosis is poor mainly due to urological (e.g., bilateral renal agenesis) and cardiac malformations.

The next group consists of patients with a **complete absence of the sacrum**. Type I, according to Pang, is a group of children with a complete absence of the sacrum and the absence of several lumbar bodies (anchipod type according to Duhamel). Type II is the complete absence of the sacrum, but the lumbar vertebrae are present [7]. In both of these groups, ilia may be fused together, articulate with each other or with the lowest vertebra. Spinopelvic kyphosis is present and, sometimes, a scoliosis. There may be abnormal mobility at the junction of the spine with the pelvis, a “pseudo-joint”.

The transverse dimension of the pelvis is often smaller, the pelvis is narrow and the buttocks are flat. Lower limbs are in a “Buddha” position with severe flexor contractures: flexed in knee joints with popliteal webbing and abducted hip joints (Figure 2 and Figure 3). Neurogenic foot deformities in the form of clubfoot are common. Motor impairment generally corresponds to the last vertebra present, and sensory impairment may be in patches [45]. Due to a total paralysis of the lower limbs, these patients are usually confined to a wheelchair. At home, with lower limbs flexed and a relatively short torso, some of them move on their hands. The exception is the ability to ambulate at home with lower limbs in orthoses—this occurs only with the presence of all lumbar vertebrae and motor deficit from lower lumbar level [37]. Genitourinary and gastrointestinal anomalies are common, with neurogenic bladder and bowel incontinence.

The third and probably the most diverse group of patients are those with **partial absence/underdevelopment of the sacrum and coccyx**. Patients can suffer from a symmetrical or asymmetrical absence of distal parts of the sacrum. The clinical picture is diverse—from practically no symptoms in the absence of the coccyx (when the diagnosis is often made accidentally only in the teenage years) to significant neurological deficits and multi-level orthopedic problems involving lower limbs, such as a narrow pelvis, flat buttocks (poor gluteal musculature) and contractures of the lower limbs. There may be neurogenic unilateral or bilateral dislocations of the hips, neurogenic deformities of the feet, most often a clubfoot and, less often, a calcaneo-valgus. Unilateral sacral agenesis may be associated with hypoplasia of the entire lower limb.

The presence of scoliosis may be associated with vertebral defects of formation and segmentation, and may be affected by tethering of the spinal cord. Motor and sensory impairment is of lesser degree than in the previous type of CRS, but follows the same rules: paresis is usually greater than sensory impairment [7].

In addition, neurological symptoms may be unilateral, intensity may be asymmetric, depending on symmetry or asymmetry of sacral agenesis. Slight partial sacral agenesis is most common; therefore, paresis affects muscles innervated by the S2–S5 cord levels (muscles of the perineum and pelvic sling and intrinsic muscles of the foot) [45]. This results in dysfunction of the urethral and anal sphincters and, sometimes, sexual dysfunction in males. The foot is usually drooping, cavus, with claw toes.

Patients with spinal cord abnormalities at a slightly higher level: L5–S1 (subtotal or complete sacral agenesis) have more severe neurological symptoms. The lower limbs are hypoplastic, tapered legs—the thigh is properly built, and the further parts of the limb with muscle atrophy in the posterior and sagittal group of the calf, and with foot deformities. The defect may affect only one limb in the case of significant asymmetry of the sacrum (Figure 4). Neurologic bladder and fecal incontinence may be observed.

To sum up, the presence of visible defects in the spine and lower limbs as well as imperforate anus results in early diagnosis of the defect, immediately after birth. Less severe defects with slight underdevelopment of the sacrum may be diagnosed later in life. The most common deformities of the spine and the central nervous system include failure of formation and segmentation, scoliosis and abnormal kyphosis, tethering of the spinal cord and its abrupt termination [10]. More than 80% of patients with CRS have limb defects and other non-vertebral bone defects. Limb shortening/hypoplasia, club feet, popliteal webbing and contractures are the most common orthopedic extra-spinal problems [10].

## 6. Urogenital and Gastrointestinal Problems

Most CRS defects are diagnosed immediately after birth—this applies mainly to more severe forms of the defect. In less severe sacrococcygeal agenesis, the diagnosis may be made only in children who are a few years old [11]. Nevertheless, early diagnosis is crucial due to the risk of neuropathic bladder [3,11]. Delayed diagnosis may lead to an increased risk of recurrent urinary tract infections, incontinence and even renal impairment. The absence of more than one vertebra in the sacrum can lead to neuropathic bladder—although, most often, it is a defect between S2 and S4 [11]. Neurogenic bladder is a neurological problem, but these patients may also have genitourinary anomalies: nonspecific hydronephrosis, renal dysplasia/agenesis (or ectopia), dysplasia or agenesis of other parts of the genitourinary system [4,6,19,49]. An unfavorable factor is vesicoureteral reflux, which, together with frequent urinary tract infections, may affect kidney function in the future. Sinha, in his interesting research, observed a long-standing lower urinary symptoms in all patients with sacral agenesis, as well as high prevalence of upper tract changes [50]. Isolated sacral agenesis may be a cause of neurogenic bladder that often presents itself late and may result in renal damage [50]. Esposito recommended a special diagnostic protocol for all newborns at increased risk of urologic anomalies. It consists of performing an ultrasound examination in all patients from this group and, if any abnormalities are detected, conducting a urodynamic or videourodynamic study [3]. Gastrointestinal anomalies can lead to problems with bowel movements and control (incontinence and encopresis). Fecal incontinence, according to some studies, concerns 1/3 of patients, and imperforate anus is a significant risk factor [3,51].

We must consider that sacral abnormalities range from missing the coccyx to complete absence of the sacrum with fused iliac bones [52]. The diagnosis is not always established in the first months of life. That is why sacral abnormalities should be suspected in patients with early severe diaper rash and failure to toilet train as the early symptoms of fecal incontinence and neurogenic bladder [52].

## 7. Treatment

Treatment of patients with CRS is complex and multidisciplinary and should be comprehensive. The severity of the defect varies; hence. In various medical problems, the management depends on the specific anatomical abnormalities present in a given patient [53]. Singh described it well in one sentence: “the treatment is challenging for the treating physician as well as for the parents and calls for a multidisciplinary approach” [6]. The primary damage is irreversible, only “repair” treatment for each individual system remains. The most important are treatments of bladder and bowel continence, preservation of renal function and orthopedic deformities [6,54]. Early surgical treatment mainly concerns tracheoesophageal fistula, cloacal anomaly, omphalocele, bladder exstrophy, imperforate anus and other life-threatening defects [12].

Neurosurgical treatment may be necessary in patients, as defined by Lee, with the ‘failure of regression’ type [12]. In this group of patients, unlike the “failure of formation” type, there is a risk of worsening neurological damage due to the tethering of the spinal cord [7,12]. For patients with spinal cord tethering lesions, release of the conus and resection of selective myelodysplastic lesions was indicated [7]. Myelomeningocoele, which sometimes accompanies the defect, also requires surgical treatment in the early period of life [7,55]. A new, quite promising solution is the combination of growth hormone therapy and rehabilitation which, in early stages of life, seems to be useful for acquiring innervation of distal spinal cord segments followed by improvement the quality of life in some cases of CRS [56].

Orthopedic treatment can be divided into two areas: the spine and lower limbs (contractures, foot deformities, neurogenic dislocation of the hip). The decision regarding surgical treatment in patients with CRS should primarily consider the ambulatory potential and spinopelvic stability [47,57].

### 7.1. Spine

Two spinal problems may occur in patients with CRS spinal deformity (kyphosis and scoliosis) and spinopelvic instability due to the incorrect connection of these structures [47,54]. In the absence of significant pelvic obliquity, bedsores and significant progression of deformities in patients classified as nonambulatory (wheelchair bound indoors and outdoors), only a conservative treatment is recommended (Figure 5) [6,58].

These patients use the mobility of the spine in everyday activities (e.g., when moving to and from a wheelchair), and spinal fusion surgery would not improve their quality of life; they may experience trunk stiffness during activities that they previously performed independently without problems.

However, surgical treatment should be considered in patients with progressive spinal deformity and spinopelvic instability—especially in ambulatory patients [57]. Spinal and pelvic stabilization with correction of spinal deformities can improve, in those cases, the trunk balance and facilitate ambulation. In the case of wheelchair-bound patients with a significant oblique position of the pelvis, surgical treatment may improve the comfort of sitting.

The spine should also be examined for atlantoaxial instability or congenital anomalies in cervical spine and adequate treatment should be applied if necessary [47]. Sacral agenesis may be also accompanied by myelomeningocele, which makes the treatment of spinal deformity even more challenging [59].

Scoliosis is a common defect associated with lumbosacral agenesis, although no correlation was found between the two [6]. Spinal curvature may result from congenital defects of the vertebrae, but it may also occur in the absence of failure of formation and segmentation. In the case of progressive deformities, surgical treatment is recommended.

Spinopelvic instability can affect walking ability and can occur with or without spinal deformity. Spinopelvic fusion is considered controversial by some authors, but most agree that in selected cases, the patient may benefit from this surgery [24,47,57,60,61,62]. Balioglu suggested posterior instrumentation and stabilization for progressive spinal deformities and lumbopelvic instability in patients with CRS, especially in the group without concomitant myelomeningocele [55]. The most common problems related to spine surgery were: implant failures, excessive bleeding, delayed wound healing and dural tears [55]. Missing even a minor defect from the sacral agenesis group in patients with congenital lumbosacral deformities is a risk factor for postoperative coronal imbalance. In such cases, sacropelvic stabilization with sufficient bone grafting at sacroiliac joint is important [63].

Spinopelvic kyphosis and instability affect sitting position, where hand support is necessary and ribs are touching the iliac crest. Griffet, in such a case, performed a spine distraction using an Orthofix external fixator, but without achieving spinopelvic fusion [64]. Yazici corrected kyphosis with posterior lumbopelvic instrumentation and fusion in three cases with tibial bone grafts [62]. Various implant systems and techniques were used by Balioglu, with pedicle subtraction osteotomy when needed [55].

Achieving stability and fusion in spinopelvic surgery may be challenging, due to a relatively high incidence of bone nonunion [1,65]. In order to eliminate this complication, various types of spinopelvic fixations were used [62,64,65,66,67,68]. In addition, it is necessary to use auto or allogenic bone grafts, even vascularized grafts [61,64,65,66]. Ferland, despite the use of vascularized rib grafts, observed seven revision surgeries in four patients [65]. A technique described by Vissarionov seemed to be more effective. It allows for correction of lumbosacral kyphosis with spinopelvic stabilization. He achieved 100% fusion rate, although surgery was performed mostly in children younger than 3 years of age and the follow-up was not extended until skeletal maturity [61].

Modern instruments allow for a withdrawal of the Galvestone technique or its modifications. The use of S1 screws, iliac screws or S2-alar-iliac (S2AI) significantly strengthens the construction of instrumentation [57,66,69]. The use of S2AI screws as distal fixation seems to be the most effective with the lowest risk of implant-related complications [69]. Despite new implants, the risk of complications in these procedures is still higher than in classical spine surgery. The most common complications include a delayed postoperative wound healing and implant loosening; however, the frequency of these was decreasing in recent reports [1,69]. Another surgical solution was proposed by Mathews in the case of severe spinal deformity with sacral agenesis accompanied by thoracic insufficiency syndrome. An expansion thoracoplasty with vertical expandable prosthetic titanium rib (VEPTR) placement was performed with improvement of pulmonary function [70].

Nowadays, the surgery in spinopelvic deformity and instability is a relatively safe and effective procedure that enables improvement of motor activity and verticalization of patients (depending on their neurological status) (Table 2) [61,69].

### 7.2. Lower Limbs

#### 7.2.1. Knee

Apart from the neurological condition, knee flexion contractures are a major determinant of walking ability. They can vary in severity depending on the level of agenesis in CRS: the most severe are from the first lumbar and above [54,57].

Out of contractures in the lower limbs, knee flexion contractures with popliteal webbing were the most difficult to correct. Knee flexion contracture correction is indicated with preserved quadriceps function and with potential walking ability (independent ambulators or household ambulators) [47,54]. In the case of surgery, the correction of the contracture should be complete, due to the risk of recurrence of the deformity [47].

In patients with quadriceps paralysis, there is a significant risk of recurrence of the deformity, and usually the operation will not improve the patient’s quality of life. In non-ambulatory patients with severe lower limb contracture, some authors recommended amputation or knee disarticulation to improve sitting comfort [54,58]. Renshaw even argued that amputation may be the treatment of choice for severe contractures and deformities of the lower limbs [45]. However, the authors of this paper agree with Guille et al. that it is generally not necessary and poorly accepted by both the patient and family [47]. In patients with high spinal cord failure, severe knee flexion contractures with popliteal webbing set the limbs in a “cross-legged sit”, which allows comfortable support when sitting in a wheelchair and allows patients to move at home on their hands with lifting the entire torso; it also provides a slightly better cosmetic effect compared to amputation.

#### 7.2.2. Hip

Hip flexion contractures and their dislocations are another problem in the lower limbs in patients with CRS (Figure 4A) [47,54,71]. There is no relationship between the degree of hip dysplasia and the severity of CRS [47]. A good prognosis regarding the possibility of upright positioning and walking determines indications for surgical treatment. Patients from group A, according to Guille, i.e., those who walk or could walk independently, should be treated surgically. In the remaining groups of patients, surgical treatment is indicated only exceptionally if the position of the limb interferes with sitting or orthotic fitting [55,58]. Surgery in hip dislocation should be performed early and according to usual rules (closed reduction, open reduction, osteotomies, if necessary) [47]. In cases of neurogenic hips with disturbed muscle balance, and this group includes hip joints in CRS, the subluxation/dislocation easily recurs as the child grows. It should be taken into account that additional surgeries may be necessary during adolescence.

#### 7.2.3. Foot

Foot deformities are common among patients with CRS, including flexible as well as rigid clubfoot or equinus contracture [47,53,54,58,72]. Balioglu reported that up to 63% of patients with sacral agenesis suffered from foot deformities, with the most common being clubfoot deformity (Figure 6) [55].

Calcaneovalgus foot deformities are much less common [46,57]. In patients with partial sacral agenesis, the spectrum of foot defects may be much larger—from clubfoot and valgus foot to calcaneus and cavus foot [73,74]. Foot deformities may be accompanied by contractures and clawed toes. The image of the feet is similar to the deformities found in patients with myelomeningocoele and spinal dysraphism [74,75]. In patients with tethered cord syndrome, foot alignment may deteriorate with age with increasing neurological deficits. This may signify the first symptom suggestive of tethering; therefore, worsening foot deformity should prompt an urgent diagnosis and neurosurgical consultation. Due to the concomitant sensory disturbances and soft tissue trophic disorders, there is an increased risk of ulcers that are difficult to heal when the foot is loaded incorrectly. Such non-healing ulcers may require debridement and reconstructive surgery, with lateral supramalleolar flap, as described by Yamamichi [76].

The general principles of treatment of foot deformities do not differ from the treatment of other deformities of the lower limbs—ambulatory patients require treatment as a rule, while non-ambulators only if there are risk of pressure sores or problems with orthotic fitting. The goal of treatment for ambulatory patients is to achieve a painless, plantigrade foot [71]. If only plantar contracture is present, gastrocnemius lengthening or posterior release or tenotomy may suffice. Clubfoot deformity often resembles the foot in arthrogryposis and requires more complex treatment. In small children, we start treatment with manipulation and serial casting, according to Ponseti [73,77]. Ineffective conservative treatment is the indication for surgery—from soft tissue release to bone procedures (osteotomies and arthrodeses) [47]. Bray described an instructive case of a girl with clubfoot on the right and vertical talus on the left. First, the foot deformities were treated typically (with the Ponseti method), but when child failed to achieve developmental milestones and examinations revealed abnormal lower limb reflexes, the diagnosis of sacral agenesis was established. At the end, due to resistant deformity, both feet required open surgery before the first year of life [73].

In valgus feet, as in cerebral palsy, talocalcaneal arthrodesis is used (e.g., according to Grice). Although triple fusion is currently being abandoned in favor of joint-sparing techniques (various osteotomies), in patients with CRS, this procedure may be a good solution, especially in large and/or recurrent deformities [47]. In non-ambulatory patients, surgical treatment of the feet is indicated only in selected cases. Formation of bedsores and trophic ulcers due to incorrect positioning of the foot on the footrest of the wheelchair or inability to put on shoes (especially in winter) may be the indication for surgery when conservative treatment fails [76]. The operated foot should “fit” the shoe and should not cause irritation or reddened places signifying the risk of pressure ulcers. The surgical procedures involve similar techniques, as in the case of ambulatory patients, and each intervention is adjusted to individual patient’s need.

## 8. Discussion

Our study, which is a review of the available literature, with particular emphasis on the last 10 years, showed the enormous complexity of the problem. It should be noted that most of the publications are case studies, which, to some extent, makes it difficult to establish treatment standards.

The treatment of CRS patients is a complex problem and a “never-ending story”. Each case is different and should be considered individually (Figure 7).

Correct diagnosis of the defect, with all its components, should be made as soon as possible, even in prenatal period, to allow for timely planning of the appropriate treatment which is primarily focused on the treatment of problems that directly threaten patient’s life.

Patients with CRS usually have normal cognitive abilities, hearing and speech [78,79]. The goal of treatment is functional improvement and achieving the best possible quality of life. The main problems—urological, gastrointestinal and orthopaedical—often need extensive, sometimes multistage, surgery to prevent further complications such as progressive renal damage [78,80]. Typical for CRS, orthopaedical problems such as hip dislocation, limb deformities, spine instability, joint contractures significantly worsen the quality of life making it difficult to function in society. The goal of orthopaedical treatment is to restore mobility (even to a limited extent), including standing. comfortably sitting, and avoid amputations [55]. The severity of the syndrome can limit the range of positive outcomes, but we must also consider the possibility of worsening of the symptoms relative to patients age (often requiring additional surgical procedures) [81].

The main limitation of the study was the subjective choice of papers considered. However, on the other hand, this particular choice may also be a strength of this work since they were written by orthopedists for other orthopedic surgeons.

## 9. Conclusions

Comprehensive, multidisciplinary long-term treatment should be started as early as possible. A multidisciplinary medical team providing complex treatment should be the gold standard. Constant rehabilitation care, with cooperation with the family, is crucial.

## Figures and Tables

**Figure 1 children-10-00589-f001:**
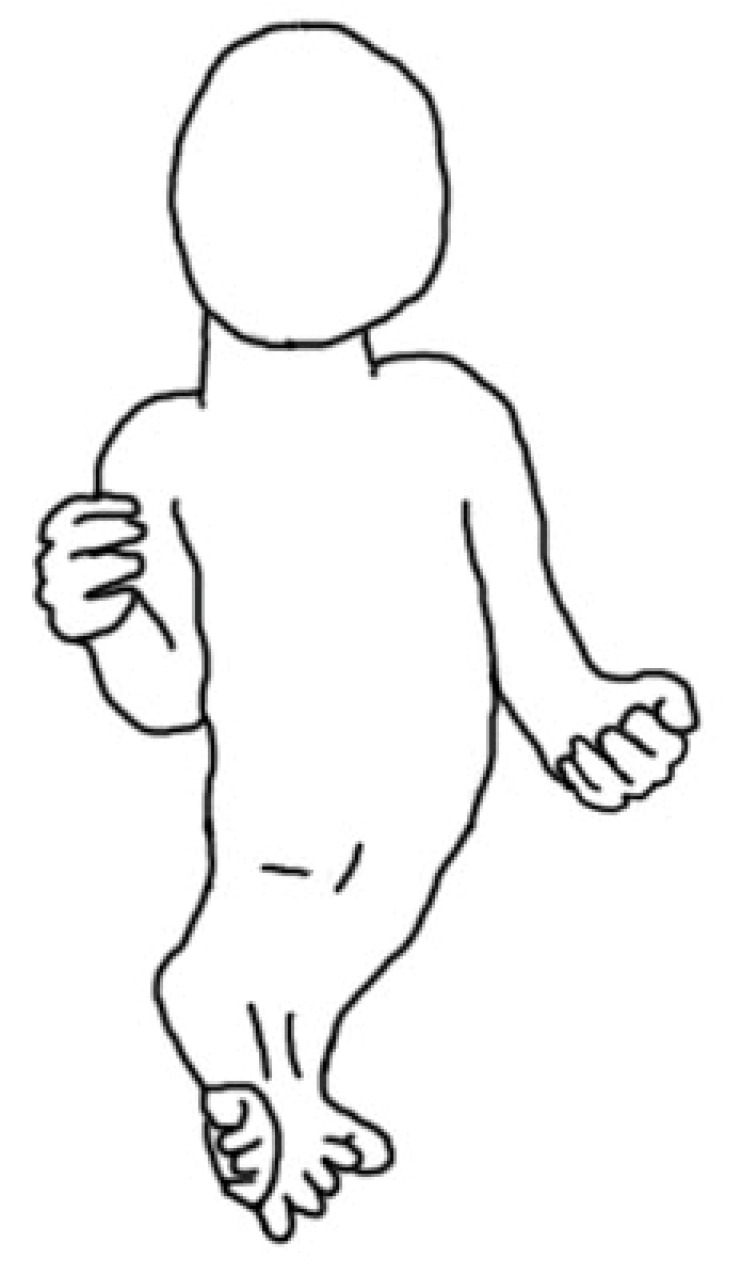
Sirenomelia.

**Figure 2 children-10-00589-f002:**
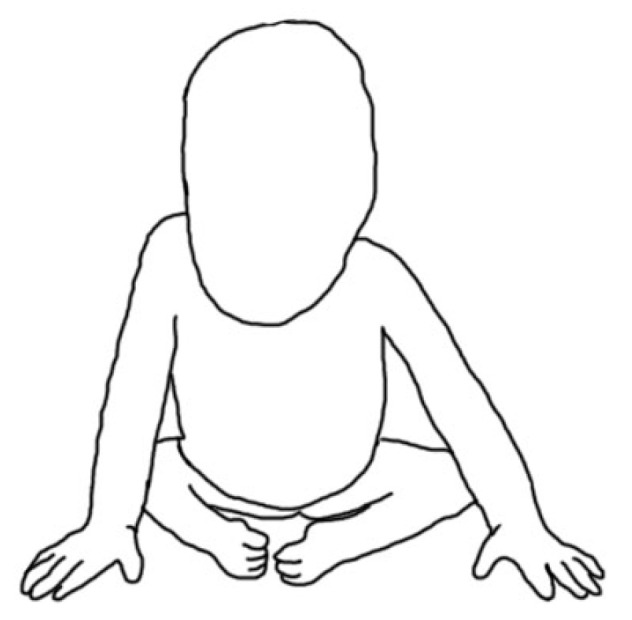
Typical sitting position of a child with complete absence of sacrum and some lumbar vertebra.

**Figure 3 children-10-00589-f003:**
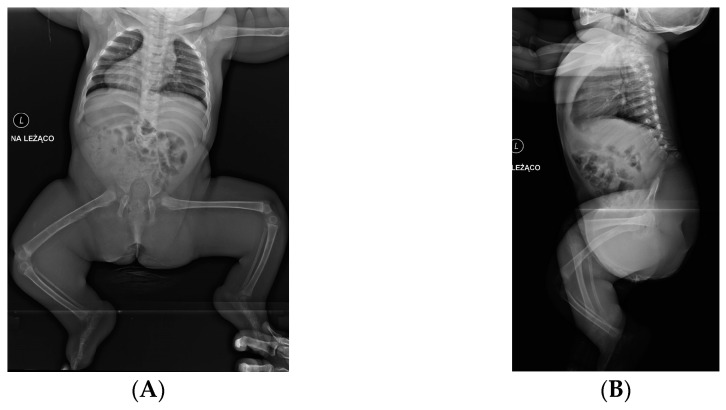
8-month-old girl with complete absence of sacrum. (**A**)—ap view, (**B**)—lateral view.

**Figure 4 children-10-00589-f004:**
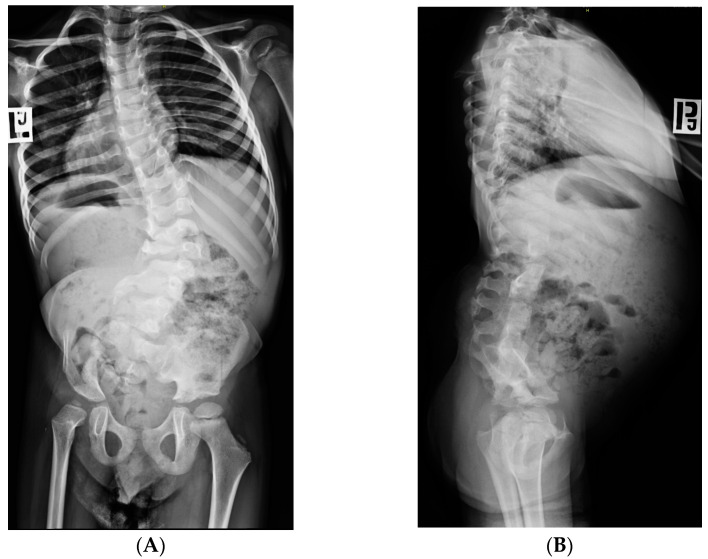
A boy with partial absence of sacrum, scoliosis and hip dysplasia. (**A**)—ap view, (**B**)—lateral view.

**Figure 5 children-10-00589-f005:**
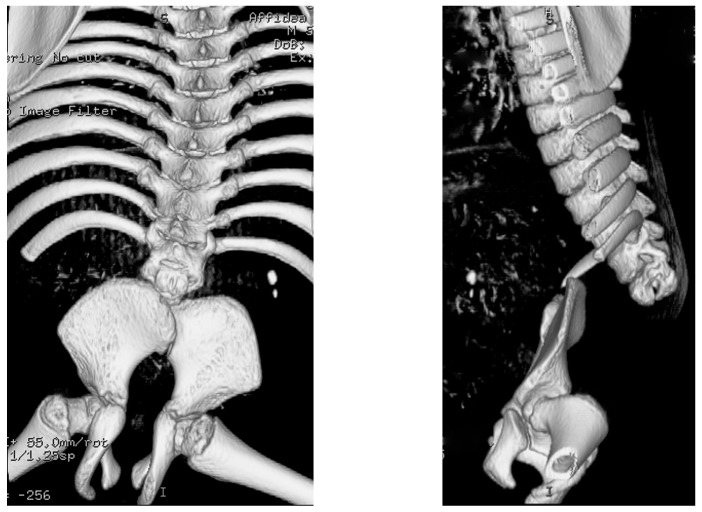
CRS. CT scans of spine and pelvis in total agenesis of sacrum and lower lumbar vertebra.

**Figure 6 children-10-00589-f006:**
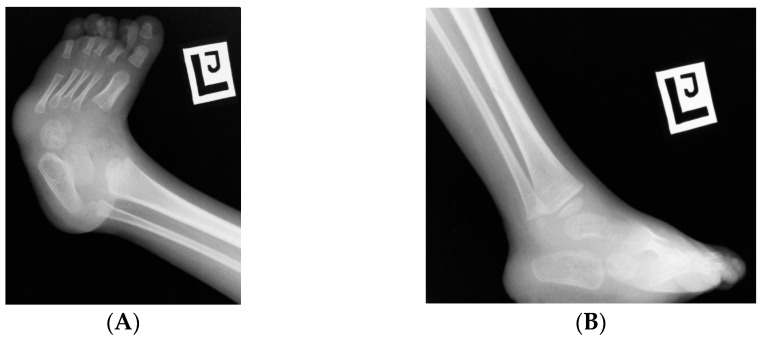
(**A**,**B**) Neurogenic clubfoot deformity. Partial sacral agenesis.

**Figure 7 children-10-00589-f007:**
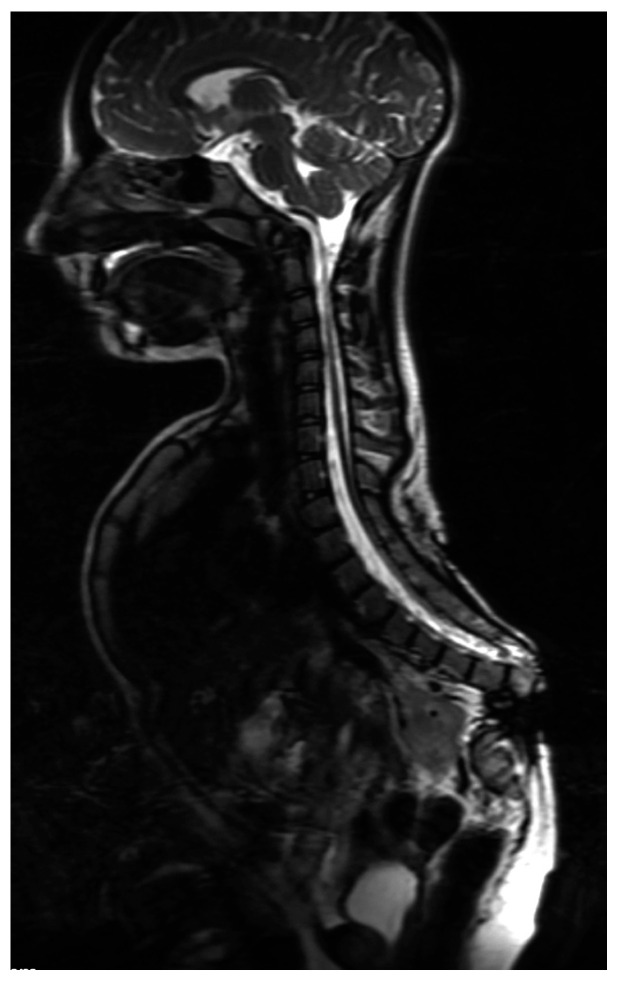
MRI scan. Atypical CRS: absence of lower lumbar vertebra and sacral hypoplasia.

**Table 1 children-10-00589-t001:** Number of papers found in PubMed and Google Scholar using specific key words.

	PubMed	Google Scholar
“caudal regression syndrome”	395	3390
“sacral agenesis”	415	6000
“sirenomelia”	2588	3050
“caudal regression syndrome” AND “treatment”	118	2850
“caudal regression syndrome” AND “surgery”	96	1700
“sacral agenesis” AND “treatment”	197	4200
“sacral agenesis” AND “surgery”	184	4100
“caudal regression syndrome” AND “spine”	176	1650
“sacral agenesis” AND “spine”	239	2870
“caudal agenesis”	2857	23,400
“caudal agenesis” AND “treatment”	1030	17,500

**Table 2 children-10-00589-t002:** The most interesting papers on spine in caudal regression syndrome, published in recent years.

Authors and Title	Year of Publication
Balioğlu MB et al.: Sacral agenesis: evaluation of accompanying pathologies in 38 cases, with analysis of long-term outcomes [55]	2016
Ferland CE et al.: Bilateral vascularized rib grafts to promote spinopelvic fixation in patients with sacral agenesis and spinopelvic dissociation: a new surgical technique [65]	2015
Griffet J et al.: Lumbopelvic stabilization with external fixator in a patient with lumbosacral agenesis [64]	2011
Mathews CS et al.: Expansion Thoracoplasty as a Life-Saving Procedure in an Adolescent With Severe Spinal Deformity and Sacral Agenesis [70]	2019
Szumera E et al.: Atypical caudal regression syndrome with agenesis of lumbar spine and presence of sacrum—case report and literature review [1]	2018
Vissarionov S et al.: Surgical Correction of Spinopelvic Instability in Children With Caudal Regression Syndrome [61]	2019
Yazici M et al.: Lumbopelvic fusion with a new fixation technique in lumbosacral agenesis: three cases [62]	2011
Zhang H et al.: Sacral agenesis combined with spinopelvic dissociation: A case report and literature review [57]	2018
Zhang T et al.: Different distal fixation anchors in lumbosacral spinal deformities associated with sacral agenesis: which one is better? [69]	2021
Zhang T et al.: Sacral Agenesis: A neglected deformity that increases the incidence of postoperative coronal imbalance in congenital lumbosacral deformities [63]	2022

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
