# Peer review of "Caudal Regression Syndrome—A Narrative Review: An Orthopedic Point of View"

_children, 2023, doi:10.3390/children10030589_

Round 1
Reviewer 1 Report
It is a well written work. The subject is interesting and is not frequently addressed.
It provides an integrative approach that provides an updated and interesting vision for professionals.
The introduction is appropriate, describes the target pathology well, and provides clear rationale and objectives.
The methodology is not described, as it is a narrative review type of work. In the case that requires revision, it is suggested to consider the inclusion of a brief materials and methods section where the authors can describe in better detail the methodology used to choose bibliographic sources, such as search engines used, search keywords, and selection criteria. of articles (language, type of work, accessibility, etc.). This would provide a high quality plus to the manuscript.
The main body of the work is well developed, and sequentially addresses the different variants of the syndrome, providing a very interesting integration of bibliographic evidence with clear experience of the authors.
There is no clear section for discussion and/or conclusions, which I think should be re-evaluated by the authors. A brief discussion should be included where it is detailed: similarity of the work with others, existing differences between local experiences, and strengths and weaknesses of the present work, with a description of its own limitations (which will surely be linked to the methodology and limitations of the bibliography). used).
The figures are very interesting and clearly illustrative, although the graphic quality is what is usually expected. I suggest using them in a reduced size (as long as it is a line drawing) or evaluating if they can be replaced (at the discretion and possibility of the authors) by photos and/or more defined figures.
Author Response
Dear Editors and Reviewers,
Thank you for your comments.
We are sending the second version of the manuscript ” Caudal regression syndrome- a narrative review. An orthopedic point of view”.
We did our best and tried to change the manuscript according to your suggestions:
- X-ray and MRI pictures of the deformity were added to manuscript.
- Molecular basis and genetics studies were mentioned
- Latest trends in the management of CRS were described with more details.
- methodology was described in the section: “material. methods”
- brief discussion was added
- the paper is longer, with more figures and tables, as well as more references.
- the language was checked by our friend – native speaker (USA).
We hope our explanations will be sufficient.
Best regards,
Barbara Jasiewicz
Wojciech KÄ…cki
Reviewer 2 Report
1. The manuscript can be enhanced by providing X-ray and MRI pictures of the deformity.
2. Molecular basis as evidenced in preclinical animal studies should be mentioned.
3. Latest trends in the management of CRS should be mentioned.
Author Response

(The authors gave the same response as above.)

Round 2
Reviewer 2 Report
Thanks.